# Don't forget private retrieval: distributed private similarity search for large language models

## Abstract

While the flexible capabilities of large language models (LLMs) allow them to answer a range of queries based on existing learned knowledge, information retrieval to augment generation is an important tool to allow LLMs to answer questions on information not included in pre-training data. Such private information is increasingly being generated in a wide array of distributed contexts by organizations and individuals. Performing such information retrieval using neural embeddings of queries and documents always leaked information about queries and database content unless both were stored locally. We present Private Retrieval Augmented Generation (PRAG), an approach that uses multi-party computation (MPC) to securely transmit queries to a distributed set of servers containing a privately constructed database to return top-k and approximate top-k documents. This is a first-of-its kind approach to dense information retrieval that ensures no server observes a client's query or can see the database content. The approach introduces a novel MPC friendly protocol for inverted file approximate search (IVF) that allows for fast document search over distributed and private data in sublinear communication complexity. This work presents new avenues through which data for use in LLMs can be accessed and used without needing to centralize or forgo privacy.

## 1 Introduction

Heavily pre-trained and fine-tuned Large Language Models (LLMs) have demonstrated exceptional performance on zero-shot (Kojima et al., 2022) and few-shot tasks (Brown et al., 2020). The ability of these models to generalize, combined with their costly pretraining, has shifted the focus from training ad-hoc models to perform specific tasks to utilizing these general-purpose foundational models for a wide variety of use-cases (Eloundou et al., 2023; OpenAI, 2023). These pre-trained models lack knowledge of private contexts or recent events.

To provide these LLMs with up-to-date or relevant information, methods such as Retrieval Augmented Generation (RAG) (Lewis et al., 2020; Karpukhin et al., 2020; Mao et al., 2020) are used to include external information into a generation process without needing fine-tuning on new data. This process allows LLMs to first query an external data source, retrieve relevant information (with respect to a given prompt), and then use both the prompt and the retrieved data as input to the inference phase of the LLM.

Similar to the problem of federated learning (Kairouz et al., 2019), it is valuable to aggregate sensitive data from multiple (perhaps many) data owners. To do that, each party should be able to guarantee that their own private data remains private even when it is utilized. On the other hand, model users should be able to query these data from many data owners without needing to share what questions they are asking.

In this work we argue that LLMs require a new model for sharing data for AI tasks. Compared to federated learning, which focuses on the training phase, LLMs should focus on the (i) retrieval phase; (ii) inference phase. For the latter, trusting an external provider (e.g., OpenAI) may suffice for many. Alternatively, secure inference solutions (Li et al., 2022; Dong et al., 2023; South et al., 2023; Mo et al., 2020) may provide a solution.

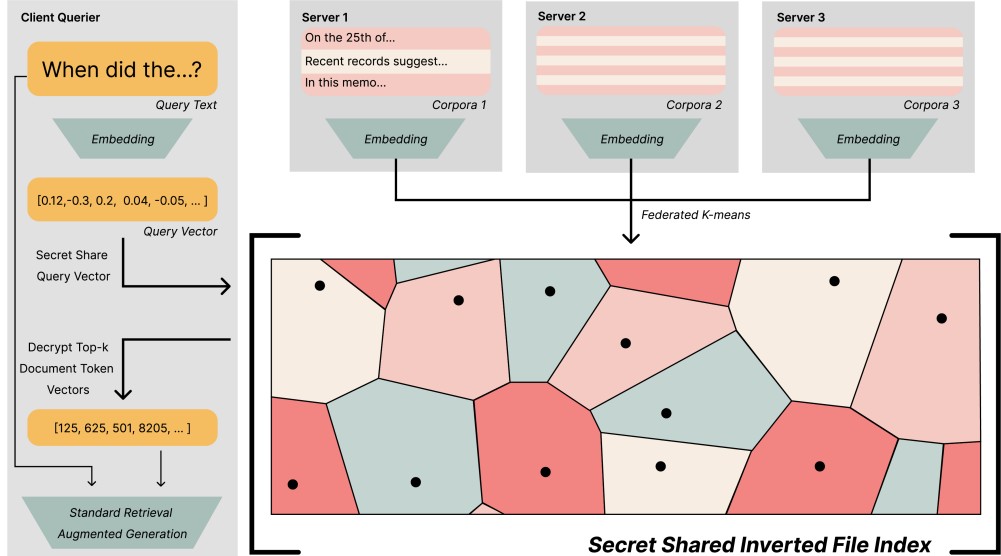

Figure 1: Overview of PRAG architecture using a distributed, secret-shared inverted file index (IVF), for retrieving document token vectors closely matching a privately-generated query vector in LLM-based question answering.

With respect to the retrieval phase, to the best of our knowledge, we are the first to offer a solution, which we call Private Retrieval Augmented Generation (PRAG).

**Our approach and contributions.** In this paper, we propose Private Retrieval Augmented Generation (PRAG), a secure approach to augment neural information retrieval that hides both query vectors and the retrieval database. We use a retrieval database split across a set of servers, and we ensure data remains private by using secure multi-party computation (MPC) techniques. To the best of our knowledge, we are the first to consider the problem of secure distributed retrieval in the context of LLMs, and more broadly, are the first to propose a solution for private similarity search that can protect both the query and a database constructed by multiple data owners across. This approach can be deployed with any standard neural IR embedding model to augment distance calculations (e.g., cosine, dot, euclidean) and top-k retrieval over federated vector stores, scaling to medium-size databases with very little accuracy loss (99% accuracy on real data).

We further scale the approach to much larger databases using an approximate k-nearest-neighbors approach inside MPC, replicating the accuracy of the state of the art in approximate retrieval using a first-of-its kind inverted files index inside MPC, providing significant speed improvements for retrieval. Our approach provides both theoretical and empirical improvements of value. We achieve constant communication on the client's side and *sublinear* communication on the servers' side — the bottleneck in MPC approaches. This work is the first IR approach to work across more than two servers with minimal additional costs. We further present a 'leaky' version of the protocol that allows for partial privacy of queries under a privacy budget with significant improvements to speed.

We evaluate PRAG across a range of data distributions, both real and synthetic, to show it broadly maintains the performance characteristics of non-secure IR approaches. We provide a pytorch-native implementation of our system using the Crypten MPC engine and retrieval hooks for langchain and BEIR.

## 2 METHODS

In this section, we present the Private Retrieval Augment Generation (PRAG) framework. The method builds from secret sharing and MPC friendly exact top-k calculations to a new MPC design of an inverted file index for efficient approximate top-k calculation.

## 2.1 Overview and Trust Model

Although a wide array of approaches exist for training document embedding models and augmenting generation with retrieved models, most neural information retrieval methods are underpinned by a step where a querier sends a query embedding to a server to calculate the distance / similarity between the query vector and the database, in order to return a document either as an embedding vector for concatenation or with the document tokens for use in LLM inference. This setup offloads the storage of large databases and their associated calculations to a more powerful server.

Recently, a significant body of research has been focusing on the problem of secure inference, which ensures that a query remains private at all times. Whether secure inference is achieved through cryptographic techniques (e.g., Li et al. (2022); Dong et al. (2023); Akimoto et al. (2023); Chen et al. (2022); Gupta et al. (2023)), or by running the model locally (Arora & Ré, 2022), if the inference pipeline includes an external retrieval phase (as is often the case), then security does not hold as the query itself is leaked to the database operator.

Similarly, the database may itself hold private information, collected by many different data owners. The only way to protect their data is by making sure both the client and the vector database server(s) remain oblivious to its content.

To formalize this, we assume our system has $n_{clients}$ clients sending queries and $n_{owners}$ data owners. Both clients and data owners interact with a set of $n_{servers}$ vector database operators. We assume that all parties in the system are semi-honest (i.e., they follow the protocol) and that at most $t < \frac{n_{servers}}{2}$ of the servers are corrupt (the honest majority setting). In this work, we do not focus on the $n_{owners}$ data owners privately building the server, and we assume that at some point in the past these data owners have secret-shared their data to the servers. Instead, we are focused on the inference stage, a much more frequent and real-time operation.

## 2.2 Exact MPC Tools

We assume all values are shared using Shamir (1979) secret sharing over a prime field $\mathbb{F}_p$ where $p \cong 32$ or 64 bits. We note that our protocols could work using other secret sharing schemes suitable for the honest-majority setting (e.g., replicated secret sharing (Ito et al., 1989) over the ring $\mathbb{Z}_{2^{32}}$ or $\mathbb{Z}_{2^{64}}$).

We further assume, as is common in secure machine learning literature (e.g., Riazi et al. (2018); Knott et al. (2021)), that there is a trusted dealer that generates shared random values. However, other techniques could distribute this (e.g., Damgård et al. (2013); Orsini et al. (2020); Escudero et al. (2020)). As in other works, since these protocols happen offline in a preprocessing phase and do not impact the online performance of serving a query, we do not benchmark their performance.

We denote arithmetic secret-shared values by $[x]$. A share for a specific server $i$ is denoted as $[x]_i$. When sharings may appear once as a $t$-degree sharing and another as a $2t$-degree sharing, we occasionally distinguish these sharings with a superscript (e.g., $[x]^{(2t)}$). We use $[x] := \text{SS.Share}(x)$ and $x := \text{SS.Reveal}([x])$ for sharing and revealing secret shared items.

As is well known, all linear operations over secret-shared values require no interaction between the servers. For multiplication, a single round of interaction is required. Given our setting, we find the multiplication protocol by Damgård & Nielsen (2007) to be the most suitable. For completeness, we briefly describe the protocol: given secret-shared values $[x], [y]$ the protocol securely produces $[z]$ such that $z := xy$. The protocol begins by having a pre-computed double sharing of $[r]^{(t)}, [r]^{(2t)}$ secret-shared between the parties. Then, each party $i$ locally computes a partial masked multiplication of $[\tilde{z}]_i^{(2t)} := [x]_i[y]_i + [r]_i^{(2t)}$. All parties use $\text{SS.Reveal}([\tilde{z}]^{(2t)})$ in a single round of communication to obtain $\tilde{z}$, and finally they each set $[z]_i := \tilde{z} - [r]_i^{(t)}$ to obtain a sharing of the result.

We note that recently Goyal et al. (2021) showed how to improve the efficiency of this protocol. Our work could leverage this optimization technique in a black-box manner without further changes.

Since in this work we operate in the semi-honest, honest-majority setting, we encode real numbers into a field, we use the common technique of representing all underlying values as fixed-point integers (Catrina & Saxena, 2010). In practice, this means that for any real value $\tilde{x} \in \mathbb{R}$, we encode

it as a fixed-point integer $\lfloor \tilde{x}2^f \rfloor \in \mathbb{Z}$ with precision $f$. Note that multiplying two encoded values results in a value with $2f$-precision. Therefore, truncation is needed after every multiplication to avoid causing an overflow inside the field, which would distort results.

### 2.2.1 DISTANCE CALCULATIONS

While there is some heterogeneity in distance measures used in neural information retrieval, the majority use dot products, cosine similarity, or L2 norms (euclidean distance). We provide MPC friendly implementations of all three.

A naive implementation of a dot product between a vector and a matrix can be provided by running the secure multiplication protocol in parallel. Both the communication and the computation complexity scale linearly with the size of the database $N$ and embedding dimension size $d_e$, the latter of which is fixed in almost all cases. Round complexity remains the same (constant) regardless.

Extending the dot product gives us cosine similarity, the predominant distance measure in sentence transformer style models (Reimers & Gurevych, 2019). To save on expensive MPC computations, we pre-normalize the input vectors and matrices prior to secret sharing into MPC, allowing for cosine similarity to reduce to a simple dot product. Computing Euclidean distance can also be achieved directly through MPC, but we observe that this is a much more expensive operation, as it requires computing square roots inside the MPC circuit. For example, Crypten (Knott et al., 2021), which we use in our implementation, uses a slow Newton-Raphson approach for computing square roots, requiring multiple rounds of communication.

However, we make the observation that given that top-k calculations are the end goal of distance calculations, the monotonic square root step in L2 can be ignored completely before looking for the top-k elements in the distance vector, removing the need to compute the square root securely.

### 2.2.2 FAST SECURE DOT PRODUCT

Computing the dot product of two vectors $x, y$ requires computing the sum of their point-wise products $z := \sum_{j=1}^{d} x_j y_j$. This can be achieved in MPC naively by using a secure multiplication protocol in parallel. However, for vectors of size $N$, this requires pre-processing and communicating $O(N)$ elements per dot product. This further compounds as we try to securely multiply matrices together, as in our case.

However, as was observed by Chida et al. (2018) and leveraged in works such as Abraham et al. (2020), we can reduce the communication complexity of computing a dot product from $N$ elements to a single element, by first having each party first locally compute the sum of point-wise products (instead of each product independently), and only masking the final result, as is shown in Protocol 1. Repeating this across a dimension of a matrix, we can use this for efficient matrix multiplication.

---

**Algorithm 1:** $\Pi_{\text{SumProd}}$

---

**Input:** Public Parameters: $t, d$
Input: $[x]^{(t)}, [y]^{(t)}$ two input vectors of size $d$ given as $t$-sharings
Preprocessed: $([r]^{(t)}, [r]^{(2t)})$
**Output:** Returns $[z]^{(t)}$

1 Compute $[z]^{(2t)} := \sum_{j=1}^{d} [x]_j [y]_j$ // local dot product;
2 Compute $[z]^{(t)} := \text{SS.Reveal}([z]^{(2t)} + [r]^{(2t)}) - [r]^{(t)}$ (Re-randomize and reduce sharing);
3 Return $[z]^{(t)}$;

---

### 2.2.3 RELATION TO PRIVATE INFORMATION RETRIEVAL

A well-known method of privately reading a specific entry in a database is by computing the dot product between a one-hot-vector with a non-zero element at the index of interest. Assuming $i$ is the index of interest from some arbitrary vector or matrix $x$, one can privately retrieve the data at row $i$, without leaking any information as $[0, \ldots, 1, \ldots, 0] \cdot [x_1, \ldots, x_i, \ldots, x_N]^T = [x_i]$. To read several rows at once, we can first sum across several one-hot-vectors to obtain a single vector.

This simple oblivious private retrieval from a database allows us to extract any top-k elements from a database matrix that has been secret shared. This allows us to extract either database embedding vectors or token arrays from inside the distributed database for return. In essence, rather than securely returning top-k indices and asking the user to separately extract them, we can return the original tokens from a secret shared database directly in MPC. This oblivious retrieval is used extensively throughout our protocols below, such as in extracting candidate vectors from clusters.

### 2.2.4 EXACT TOP-K FOR RETRIEVAL

Retrieving the most similar documents to a query requires first ranking all documents by some similarity metric (as above) and then picking the top $k$ documents that are closest to the query.

Our solution is conceptually similar to secure top-k circuits designed in other works such as Chen et al. (2020), where $O(kN)$ comparisons are needed. These circuits operate by successively keeping an ordered list of $k$ items, and then computing each value in the array with the minimum value in the (much smaller) sorted list. Unfortunately, this solution also requires $O(N)$ rounds for MPC based on secret-sharing.

Instead, our protocol iterates $k$ times over a secret-shared vector $[x]$. In each iteration, we run argmax($[x]$) to extract the current minimum's index in the vector. We then obliviously scale down the selected value enough so it will be ignored in future iterations.

There are many ways to implement an MPC protocol for argmax($[x]$). Our description above assumes a recursive tree-reduction based protocol as in Knott et al. (2021), having $O(\log_2(N))$ rounds and $O(N \log_2(N))$ total communication. This leads to an exact top-k round complexity of $O(k \log_2(N))$ and $O(kN \log_2(N))$ overall communication.

By combining this with distance calculations and oblivious private retrieval from a database we can provide an end-to-end exhaustive exact algorithm to return the top-k nearest documents to a query from a database of embeddings (and a database of tokens for exact document return).

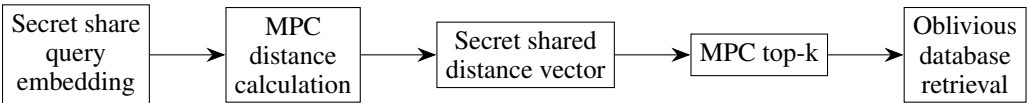

### 2.3 NEAREST NEIGHBORS AND INVERTED FILES (IVF)

At its core, the information retrieval task of top-k closest points is exactly the task of solving the *k-nearest-neighbors* (kNN) problem, which requires finding the k points in a database that are nearest to the given data point (the query). While the above exact approach achieves this, it does so at a significant speed cost (both with or without MPC), motivating the creation of approximate nearest neighbors algorithms, which only require a sublinear amount of work.

These algorithms operate by first computing a compact representation of the dataset called the *index*, and then executing queries on the index. Many approximate nearest neighbors techniques exist, and one that is particularly amenable to MPC is the *inverted files index* (IVF) (e.g., Johnson et al. (2017), Jégou et al. (2011)). This technique works by first using a clustering algorithm (e.g., k-means) over the data set to find its $n_c$ *centroids*. Then, each centroid represents a cluster holding all points associated with that cluster. In other words, this process splits the database into $n_c$ buckets.

After this one-time step, querying the data starts by computing the nearest neighbors of the query with respect to all centroids. Then, the $n_{probe}$ nearest inverted files are searched, looking for the $k$ nearest neighbors among them.

During IVF generation, parameter choices in how the index is built affect the downstream performance of the queries. We choose the number of clusters to be $n_c = \alpha \sqrt{N}$ to get sublinear complexity, where $\alpha$ is a free parameter that can be tuned. During query time, we find the distance to all $n_c$ centroids, and select the top $n_{probe}$ clusters to inspect further. As we will see during experiments, this choice of $n_{probe}$ increases the recall performance of the model, and indeed at $n_{probe} = n_c$, all clusters are inspected and the search becomes exact. However, the nature of the IVF clustering allows a smaller $n_{probe}$ to be chosen while still achieving high accuracy.

## 2.4 EFFICIENT APPROXIMATE VECTOR NEAREST NEIGHBOR SEARCH IN MPC

Bringing this into MPC, the protocol $\Pi_{\text{IVFQuery}}$ securely computes the approximate nearest neighbors using an inverted file index. The protocol assumes the servers pre-computed the secret-shared inverted index [IVF], which consists of $n_c$ lists of size $m$, both of which are of size $O(\sqrt{N})$, ensuring the overall communication complexity is sublinear. We use the MPC distance measures established above to calculate the distance between the query vector and each of the $n_c$ cluster means.

The parties then run a secure protocol of exact top $k$ as described earlier to identify the $n_{probe}$ most similar clusters. Unlike non-MPC protocols, it is critical that the servers remain oblivious as to which are the top clusters for this query. Otherwise, information about both the query and database would leak. For this reason, we require the top-k protocol to return each index as a one-hot-vector of size $n_c$ which are collectively stored in [closest buckets].

Then, the parties perform an exact-match private information retrieval to get all the vectors in the closest buckets. These [candidates] can be obliviously found through a product of [closest buckets], a mapping of centroids indices to cluster indices in the database, [IVF indices], and the entire [IVF] vector database. By obliviously reducing the entire vector database into a much smaller search space that only includes vectors from the $n_{probe}$ nearest clusters, we are able to achieve sublinear overall communication.

At this stage, [candidates] holds a reduced $(n_{probe} \times m) \times d$ vector matrix (where $d$ is the embedding dimension). [candidates indices] will similarly store the mapping from each candidate to the original database index. We proceed by running an exact nearest neighbor search again, which computes the distances between the query and all candidates and then securely gets the top-k entries. Using [candidates indices], these top-k entries are mapped back to the original database records, where documents can be obviously retrieved.

---

**Algorithm 2:** $\Pi_{\text{IVFQuery}}$

---

**Input:** Public Parameters: $n$, $k$, $n_c$, $n_{\text{probe}}$, $m$, $d$
Client: query $x \in \mathbb{R}^d$
Server: Secret-shared inverted file clusters [IVF clusters] $\in \mathbb{R}^{n_c \times d}$, Inverted file index values [IVF] $\in \mathbb{R}^{n_c \times m \times d}$, Inverted file index indices [IVF indices] $\in \mathbb{R}^{n_c \times m}$
**Output:** k-nearest-neighbors (approximate)

1 **Client computation:**
2 $[x] := SS.\text{Share}(x)$;
3 Send each server $i$ its share $[x]_i$;
4 **Servers computation:**
5 **in parallel** Iterate over [cluster] $\in$ [IVF clusters];
6     [centroid distance$_i$] := SumProd($[x]$, $[cluster]$);
7     [centroid distances] := $\{$[centroid distance$_1$]$^{(t)}$, ..., [centroid distance$_{n_c}$]$^{(t)}\}$;
8 Compute [closest buckets] := ExactTopk([centroid distances], $n_{\text{probe}}$);
9 Compute [candidates] := MatMult([closest buckets], [IVF]) and
    [candidates indices] := MatMult([closest buckets], [IVF indices]);
10 **in parallel** Iterate over [candidate] $\in$ [candidates];
11     Compute distance using SumProd and store as [candidate distances];
12 Compute [candidate top-k indices] := ExactTopk([candidate distances], $k$);
13 Compute [database top-k indices] via private exact-match retrieval of [candidate top-k indices]
    from [candidates indices];
14 Return [database top-k indices] documents via private retrieval.

---

### 2.4.1 SUBLINEAR COMMUNICATION COMPLEXITY

The client maintains an optimal communication complexity of $O(1)$, as it only needs to communicate a share of the query vector to each server.

As to the servers, in lines 5-7 a total of $n_c := O(\sqrt{N})$ elements are communicated. Computing the exact top-k over these $n_c$ distances requires $O(k \cdot \log_2(n_c))$ communication. Reducing the dataset obliviously costs $O(n_{probe} \frac{N}{m} d)$. With our choice of parameters, $n_{probe}$ and $d$ are constant, and

$m = \sqrt{N}$, yielding $O(\sqrt{N})$ communication. This gives a candidate dataset that is approximately of size $n_{probe}\sqrt{N}$. Finally, we can compute the distances and exact top-k on this reduced dataset, but as it now only contains $O(\sqrt{N})$, the overall communication of that step is $O(k \cdot \log_2(\sqrt{N}))$.

Overall, we see that end-to-end the servers communicate $O(\sqrt{N} + \log_2(\sqrt{N}))$ field elements while the client communicates $O(1)$ elements (in fact, she communicates exactly $d$ elements, as is the size of the input vector). This holds true so long as $n_{probe}$ remains small enough to be considered a constant. As the number of candidate clusters to be probed becomes $n_c$, the overall complexity of the approach becomes $O(\sqrt{N} \cdot \sqrt{N}) = O(N)$, which is no better than exact search but with additional overhead operations. Hence, $n_{probe}$ should be kept low as we will see in the experimental settings.

## 2.5 SACRIFICING PRIVACY FOR SPEED IN MPC IVF

The fast secure dot product trick above helps significantly improve the speed of the step wherein we reduce the full database to only the $n_{probe}$ clusters vectors relevant to the query. However, this step is still extremely costly, requiring the manipulation of a large database of vectors for lookup when the clusters are stored in a large matrix.

Instead, we can take an alternate approach, where each cluster is stored in its own secret shared database, with an exposed lookup table. The centroids of the database still remain secret shared and private, but during query time, the $n_{probe}$ closest clusters (shuffled to avoid exposing order) are decrypted by each server to retrieve the relevant secret shared cluster matrices, which can then be concatenated before passing into the second distance-top-k calculation. This has large speed implications, dramatically decreasing the data access time and allowing for speed more competitive with non-MPC IVF.

However, this does come at the cost of privacy. Each server will now know the $n_{probe}$ closest clusters to the query, which leaks the area in the embedding space where the query is coming from. Indeed, while the centroids are secret shared, knowing the lookup table and what a user accesses would allow an actor to determine an average point across those centroids with more queries.

To mitigate this, a query could be noised according to a privacy budget similar to differential privacy, as for sufficiently large $n_{probe}$, even a high noised query would likely contain the relevant closest clusters nearby. One slight advantage here is that larger choices of $n_{probe}$ provide more privacy (and more capacity for noising), while also increasing the overall accuracy of the search (as we see in Figure 3).

In general, this final methodological change differs from above by no longer being fully private, but is presented as part of the spectrum from slow but exact private search to fast approximate search, and finally to fastest but leaky approximate search.

## 3 EXPERIMENTS

To demonstrate the performance of these models we run a series of experiments on both synthetic and real data to determine performance properties of the implementations of these methods above.

We benchmark the retrieval accuracy and speed across a range of embedding sizes (256 to 8192), synthetic embedding distributions ($N(0, 0.05), N(0, 1), U(-1, 1)$, Binary), distance functions (cosine, dot product, euclidean), top-k values, IVF parameters, and database sizes. We perform MPC experiments on a single 2.2GHz Intel Xeon Silver CPU using Crypten's built-in communication code to spawn processes for each server.

Further to this, we test the approaches on retrieval of real neural embedding datasets from BEIR (Thakur et al., 2021) using the same environment. While there are several parallelization improvements that can be made locally within each server, our implementations of each algorithm above remain unoptimized.

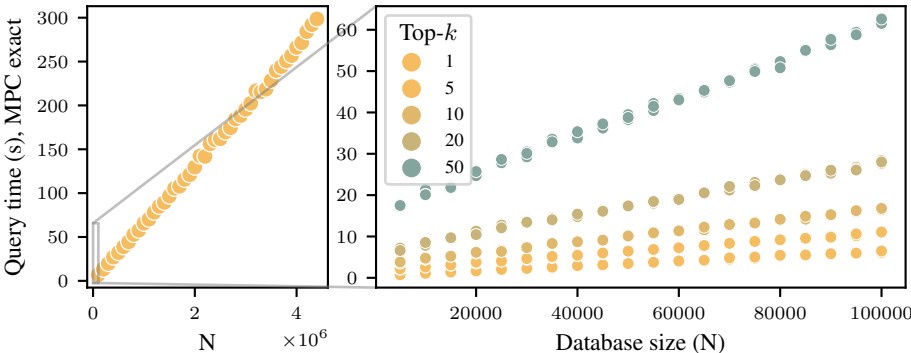

Figure 2: Time taken to retrieve top-k closest vectors in the database for end-to-end MPC exact search across increasing synthetic database sizes.

## 3.1 EXACT SEARCH

Each step of the exact search approach is extremely accurate, with small numerical errors introduced during MPC. For distance measures, MPC vectors have a mean squared error difference from pytorch calculated distances of less than $10^{-5}$ for euclidean and $10^{-8}$ for cosine, going as low as $10^{-11}$ for euclidean distance on $N(0, 0.05)$. These errors do not change with database size, and are introduced at the numerical level of the elements.

The exact top-k approach using tree reduction applied interactive k times suffers from similar small numerical errors. For distance vectors drawn $N(0, 0.05)$, where outliers are often standalone, top-k elements are picked out with 0.99 or above recall and precision. For uniform distributions (unrealistic for embedding distance vectors) the f1 accuracy is lower for top-1 (0.842) and top-k (0.96) with recall and precision climbing for higher k. This is explained by the small distances present between the max and its nearest value when drawn from a uniform distribution, leading numerical errors to induce a loss of accuracy. Fortunately, the nature of real distance distributions means performance is high in real contexts. For small values of $k$, this approach can be relatively fast but increasing the choice of $k$ dramatically increases the time cost due to communication complexity in the interactive argmax looping.

Putting distance calculations, top-k, and oblivious retrieval together, the exact search approach in MPC can identify the top-1 (argmax) most similar vector to a query with 97.5% accuracy and top-50 with 98.6% F1 score, with accuracy independent of database sizes tested up to $5 \times 10^5$. The constraint on the use of this MPC exact approach is the speed, taking up to 10 seconds for top-1 and top-5 for a $10^5$ size database, and increasing dramatically for larger $k$ as in Figure 2.

## 3.2 APPROXIMATE SEARCH

Our MPC IVF implementation, using both fully secure and partially leaky clustering, returns the elements as the standard IVF implementation with an average of over 99% recall on both synthetic and real embedding data, with errors explained by numerical errors at runtime. For real data, we use embeddings from msmarco-distilbert-base-v3 from Reimers & Gurevych (2019). These numerical errors partly flow through from the exact search above, which is used at various points in the IVF MPC algorithm. This accuracy of the MPC IVF to non-IVF is stable across choices of $n_{probe}$ and $n_c$.

While the MPC IVF matches the recall performance of the standard IVF, the underlying approximate nature of the IVF provides tradeoffs between accuracy and speed. As shown in Figure 2, increasing the value of $n_{probe}$ increases the proportion of the full database that is inspected at query time, in turn increasing the overall runtime. The benefit of IVF is that we can achieve high accuracy for even a low value of $n_{probe}$, dramatically reducing query time at the cost of accuracy.

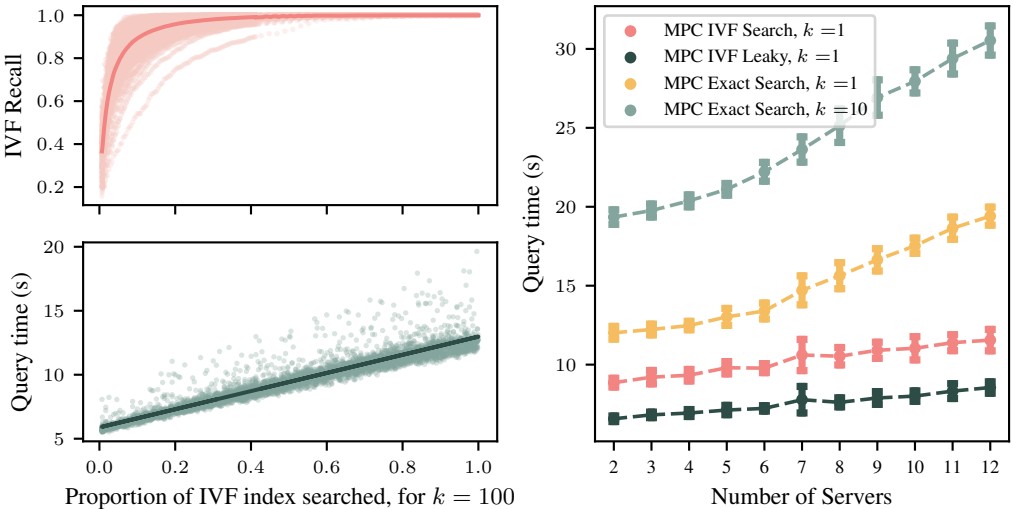

Figure 3: Information retrieval using IVF improves accuracy with increased $n_{probe}$ (top left) but increases query time as a larger proportion of the index ($\frac{n_{probe}}{n_c}$) must be searched (bottom left). These retrieval approaches (both IVF and exact) scale favorably across multiple servers (right).

## 4 RELATED WORK

Drawing on the ideas in private federated learning, we can maintain privacy when doing public queries (Arora et al., 2022) and move beyond in-context learning (Arora & Ré, 2022)/

We bring privacy to this idea through augmenting existing non-private retrieval methods, ranging from exact search on small datasets to large scale approximate retrieval (Johnson et al., 2017; Jégou et al., 2011). While several other works have examined the problem of secure similarity search (e.g., Chen et al. (2020); Zuber & Sirdey (2021); Servan-Schreiber et al. (2022); Asharov et al. (2017); Schoppmann et al. (2018); Shaul et al. (2018a;b); Songhori et al. (2015)), to the best of our knowledge we are the first to examine a model where the database is secret shared as well, and where an arbitrary number of servers and database owners can be supported. A comparison to the state-of-the-art protocols (Servan-Schreiber et al., 2022; Chen et al., 2020) is available in Table 1.

These approaches can augment other pieces of privacy-first ML infrastructure from fully secure LLM inference (Li et al., 2022; Dong et al., 2023) and federated or privacy preserving K-means clustering (Vaidya & Clifton, 2003; Jagannathan & Wright, 2005). We choose to focus on MPC techniques in this paper, as opposed to secure retrieval schemes that rely trusted execution environments (TEEs) (Wang et al., 2006; Yang et al., 2008; Papadopoulos et al., 2010; Drean et al., 2023), as TEEs have been known to suffer from privacy-breaching attacks.

## 5 CONCLUSION

We introduced PRAG, a novel approach for secure, distributed information retrieval for large language models. PRAG uniquely safeguards both query vectors and a multi-owner database using multi-party computation (MPC). Key contributions include an MPC-friendly protocol for inverted file approximate search, allowing for rapid document retrieval with sublinear communication complexity; analysis of exact search performance on language embeddings; and a version of the protocol that offers a trade-off between speed and partial privacy, under a predefined privacy budget. These tools allow for a new mechanism of neural information retrieval, which when combined with secure inference of LLMs, is a stepping stone towards fully secure foundation model agent pipelines. However, much like secure execution of LLMs, the approach put forward here has significant computational costs and speed limitations, especially for large databases and high accuracy demands. Future work should explore optimizing communication costs, enhancing protocol robustness against collusion, and integrating PRAG into larger secure machine learning frameworks.

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

## A    APPENDIX

| Protocol | Number of servers | Model | Client Communication | Server Communication | Private Database |
|---|---|---|---|---|---|
| Chen et al. (2020) | $m = 1$ | Single server | High (GBs/query) | High (GBs/query) | No |
| Servan-Schreiber et al. (2022) | $m = 2$ | Two servers (dishonest majority) | $O(\sqrt{n}log(h))$ | $O(1)$ | No |
| Servan-Schreiber et al. (2022) | $m > 2$ | Any number of servers (dishonest majority) | $O(nlog(h))$ | $O(1)$ | No |
| **This work** | $m \geq 2$ | Any number of servers (honest majority) | $O(1)$ (=input size) | $O(\sqrt{n}log(n))$ | Yes |

Table 1: A comparison of this work's contribution to distributed secure approximate kNN with previous work from Chen et al. (2020) and Servan-Schreiber et al. (2022). While Chen et al. (2020) has technically sublinear communication, it uses heavy-duty cryptographic techniques leading to higher communication costs compared to our and Servan-Schreiber et al. (2022)'s techniques.

