# OpenReview forum: "Don't forget private retrieval: distributed private similarity search for large language models"
_ICLR.cc/2024/Conference — ICLR 2024 Conference Withdrawn Submission_

### Official Review · Reviewer_zJDJ · 2023-10-30

**Soundness:** 3 good
**Presentation:** 3 good
**Contribution:** 2 fair
**Rating:** 3
**Confidence:** 4

**Summary:**

In this paper, the authors present a privacy-preserving similarity search system for large language models. Their design aims to address the privacy issue when LLM-based systems want to involve external information in the current generation process. The authors consider a scenario where the user sends queries to LLM and the LLM asks for several external server to supply extra information when generating the response. During this process, the proposed design aims to protect both users’ queries and external servers’ data privacy against the LLM owner. To achieve this goal, the authors proposed to use the Shamir’s shares, which guarantees the data privacy in an honest majority setting. The authors leverage a series of primitives based on the Shamir’s shares to compute and retrieve the top-k similarity results from an indexed database without revealing any information to compromised servers. The authors test their design with both synthetic and real-world dataset to demonstrate their system’s performance for exact and approximate queries.

**Strengths:**

1.	Address the privacy issue in the LLM augment generation process.
2.	The design provides security guarantees for both users and external information supplier.

**Weaknesses:**

1.	The design is ad-hoc, just combining existing protocols together without design efforts.
2.	The approximate search design in Sec 2.5 may lead to the leakage of centroids, which is not well-elaborated.
3.	The design is very slow and impractical.

**Questions:**

1.	It seems that such design can also be implemented with other linear secret sharing schemes, why do you choose to use the Shamir’s shares (instead of arithmetic secret shares, replicated secret shares, etc.)?

2.	The authors claim that it is possible to use perturbations to mitigate the leakage in the approximate search design, but it is unclear what the suitable parameters is for this noise.

3.	Also, it is unclear the security level after adding the noise, a proof sketch is required to understand whether the countermeasure is effectively addressing the leakage.

4.	The proposed scheme becomes very slow (at least 50s needed) when the database size goes to a million scale, but such database is quite common in the real-world. This renders the impracticality of proposed design.

5.	Even if we take the approximate design into consideration, it is still slow with a large k, I am wondering whether a large k is also common in the real-world systems because the LLM may want to embed more external resources to produce a general response for queries.

---

### Official Review · Reviewer_LqCL · 2023-10-30

**Soundness:** 2 fair
**Presentation:** 2 fair
**Contribution:** 2 fair
**Rating:** 3
**Confidence:** 3

**Summary:**

The paper introduces PRAG, which uses multi-party computation to secure the privacy of vector search. The proposed method transmits queries to a distributed set of servers containing a privately constructed database, from which it gets the approximate top-k documents. The paper is mainly about data privacy. It is more suitable to be discussed in the security or database communities.

**Strengths:**

This paper focuses on the data privacy of dense retrieval, which is a critical in practice. The multi-party computation is a meaningful method to tackle this problem.

**Weaknesses:**

This paper is entitled "distributed private similarity search for language language models". However, it literally has nothing to do large language models. Besides, the paper is neither about other seemingly related topics, like dense retrieval or text representation. The paper should be re-directed to other venues in security or database communities, which is more appropriate to justify its technical value.

**Questions:**

Please check the weaknesses.

---

### Official Review · Reviewer_Ut5B · 2023-10-30

**Soundness:** 2 fair
**Presentation:** 3 good
**Contribution:** 2 fair
**Rating:** 3
**Confidence:** 4

**Summary:**

The work studies the problem of private data retrieval for use in LLMs.
Specifically, it extends retrieval augmented generation techniques to the distributed setting where the data is assumed to be sensitive and secret shared among several servers, and the query is also secret shared.
This problem is essentially an application of $k$-NN in MPC.
The authors provide a number of solutions. The first is an exact top-$k$ computation using an off-the-shelf secure dot product protocol to compute the score, followed by an off-the-shelf argmax protocol.
The second solution uses IVF (a clustering-based technique) to reduce the runtime by only searching in specific clusters.
Finally, the fastest version uses IVF but leaks the centroid of the closest cluster.

**Strengths:**

- Very well-written paper, clear and easy to understand writing.
- An interesting new problem of protecting the privacy of retrieval in LLMs.
- While the solutions are not too novel regarding the MPC protocols they use, they present a nice variety of trade-offs in terms of runtime and privacy.

**Weaknesses:**

## Pre-computation Assumptions
The paper makes a number of bold assumptions about pre-computation:
- They assume to compute the cosine similarity that the data is normalized beforehand. If this normalization is only within each data point, that could be okay, but if the whole dataset is normalized, this would need to be done in MPC and adds non-trivial overhead.
- Similarly, the clustering for IVF would be a significant overhead in MPC that is ignored in the paper.


## Output privacy
I am concerned that, like federated learning, this technique gives a false sense of security. The paper mentions that their approach makes it possible to utilize private data for inferences. However, the privacy of the output of the MPC protocol is not considered. There needs to be at least some discussion of the threat of the output of the protocol leaking information about 1) which query was made and 2) the private data retrieved.


## Evaluation
- This paper is missing a lot of the rigour typically found in papers about MPC protocols. For example, there is no security proof or formal description of certain parts of the protocol.

- There is also no evaluation under network delay. Network delay is a realistic bottleneck of MPC that would drastically affect the runtimes in the evaluation section.

## Minor comments
- Is it realistic to consider $d$ a constant?
- Could some citations be added to 2.2.3?
- Some extra details on which truncation protocol was used would be helpful.

**Questions:**

Can the authors respond to the issues highlighted above?

---

### Official Review · Reviewer_Hhh4 · 2023-11-01

**Soundness:** 2 fair
**Presentation:** 2 fair
**Contribution:** 2 fair
**Rating:** 3
**Confidence:** 3

**Summary:**

The paper proposes a secure information retrieval method to provide retrieval augmented generation for large language models. It is an MPC-friendly inverted file approximate search protocol.

**Strengths:**

The paper proposes an interesting question of secure distributed retrieval augmentation for LLM, and introduces a framework that takes into account both query privacy and database privacy.

**Weaknesses:**

1. The experiment section does not provide a detailed introduction of the dataset, as well as the experimental settings.
2. There is a lack of comparative experiments under non-MPC settings to demonstrate the performance of the paper's method.
3. In the experiment section, the paper provides only a brief explanation of the F1 score results without presenting a complete table under different experimental settings.
4. The experiment section does not address the impact of retrieval augmentation on the performance of the Large Language Models.
5. The description of the analysis of computational complexity and communication complexity could be more detailed and standardized.
6. The full name of the abbreviation "IR" is not provided when it is first introduced.

**Questions:**

1. This paper is about secure similarity search for LLM, but it still seems to approach it from an information retrieval perspective. Please explain the relevance of the approach described in this paper to LLM.
2. Regarding risks, the paper appears to lack in-depth analysis. For example, would there be risks of model inversion attack under the scenario of multiple servers?